# Efficient Mild Organosolv Lignin Extraction in a Flow-Through Setup Yielding Lignin with High β-O-4 Content

**DOI:** 10.3390/polym11121913

**Published:** 2019-11-20

**Authors:** Douwe S. Zijlstra, Coen A. Analbers, Joren de Korte, Erwin Wilbers, Peter J. Deuss

**Affiliations:** Department of Chemical Engineering (ENTEG), University of Groningen, Nijenborgh 4, 9747 AG Groningen, The Netherlands

**Keywords:** biomass, lignin, flow-through setup, organosolv extraction, high β-O-4 content

## Abstract

Current lignin fractionation methods use harsh conditions that alter the native lignin structure, resulting in a recalcitrant material which is undesired for downstream processing. Milder fractionation processes allow for the isolation of lignins that are high in β-aryl ether (β-O-4) content, however, at reduced extraction efficiency. The development of improved lignin extraction methods using mild conditions is therefore desired. For this reason, a flow-through setup for mild ethanosolv extraction (120 °C) was developed. The influence of acid concentration, ethanol/water ratio, and the use of other linear alcohol co-solvents on the delignification efficiency and the β-O-4 content were evaluated. With walnut shells as model feedstock, extraction efficiencies of over 55% were achieved, yielding lignin with a good structural quality in terms of β-O-4 linking motifs (typically over 60 per 100 aromatic units). For example, lignin containing 66 β-O-4 linking motifs was obtained with an 80:20 n-propanol/water ratio, 0.18 M H_2_SO_4_ with overall a good extraction efficiency of 57% after 5 h. The majority of the lignin was extracted in the first 2 hours and this lignin showed the best structural quality. Compared to batch extractions, both higher lignin extraction efficiency and higher β-O-4 content were obtained using the flow setup.

## 1. Introduction

Lignin is the most abundant aromatic biopolymer and, as such, has inspired much research aiming to unlock its potential to serve as feedstock for both bulk and high-value aromatic compounds [1,2,3]. Furthermore, valorization of lignin is deemed essential for the development of economically competitive biorefineries [4,5]. However, the heterogeneous structure of lignin (Figure 1A) and the structural deviation among biomass sources has severely hampered the development of efficient catalytic routes towards selective product formation in combination with high yields [6,7]. Despite the complex structure, there are several recurring linking motifs in the native lignin structure of which the β-aryl ether motif (β-O-4) is by far the most abundant. Many of the developed depolymerization methods focus on selective breakdown of this C–O bonded linking motif, but as the majority cannot be implemented during fractionation [8], the development of methods to extract native-like lignin high in β-O-4 linking motifs is essential. Most of the applied lignin fractionation methods apply harsh extraction conditions that break a significant amount of β-O-4 linking motif [9,10,11,12,13], resulting in a more recalcitrant, condensed C–C bonded structure via repolymerization reactions, hampering the application of the selective depolymerization methodologies targeting this linking motif. In recent years, mild fractionation methods have been developed, which allow for the use of milder depolymerization methods [6,14,15,16,17]. Another applied strategy is the implementation of depolymerization with the fractionation process (lignin-first approach), like reductive catalytic fractionation which yields alkyl-phenolics in high yield but also relies on efficient initial lignin extraction [18,19,20,21,22,23,24]. One of the developed mild fractionation methods is extraction with different organic solvents which allows for obtaining lignin with relatively high β-O-4 content [25,26,27]. These extractions are performed at lower temperatures (<150 °C) compared to typical extraction methods (>180 °C). Alcohol/water mixtures are often used in these processes and yield lignin that retains most of the β-aryl ether linking motifs and is low in carbohydrate content and other impurities [6,14,28,29,30]. The presence of alcohol is important for the retention of the high β-O-4 content, as it traps the benzylic cation that is formed during the acidic extraction conditions, resulting in alcohol incorporation in the lignin structure (β′-O-4, Figure 1B) [14].

Most lignin extractions reported in the literature are performed in batch [9,11,14,30,31], but, in the last few years, the use of an increasing number of (semi-)continuous setups has been reported [32,33,34,35,36,37,38,39]. One of the major advantages of the flow-through setup is the increase in extraction efficiency compared to the batch system. Recent kinetic analysis on delignification showed that the rate determining step of the delignification process is the diffusion of dissolved lignin molecules within the cellulose layers [19,23,40,41]. The diffusive flux remains high in a flow-through setup and the lignin concentration is relatively low, as fresh solvent is constantly added to the system. In a batch system, the lignin concentration is increasing during the extraction, resulting in a decrease in diffusive flux as a function of time [23]. Other benefits of flow-through setups include shorter product residence time, more data points per experiment, and fewer scalability issues, whereas the main downside is the higher solvent consumption in flow-through setups (Figure 1B) [21,41].

Most of the reports on lignin extraction by flow-through setups use aqueous acidic solvents. The first reports focused primarily on hemicellulose hydrolysis [42,43], although the positive influence of flow rate on delignification is briefly mentioned in the latter. In recent work, flow-through fractionation of biomass using 72 wt % aqueous formic acid was performed by Wang et al. [36], showing similar lignin yields compared to batch extractions but with a significant increase in β-O-4 linking motif preservation. Comparable flow-through extractions performed at higher temperature showed a large decrease in the β-O-4 content [33,44], showing the requirement of mild conditions for the preservation of the β-O-4 linking motif. Continuous organosolv extractions have mainly been performed by applying harsh extraction conditions. Barth [39] reported 86% lignin extraction efficiency for a 10 h organosolv extraction at 175 °C with subsequent solvolysis yielding mainly methoxy-phenols with nearly 90% yield. Labbé et al. [38] used a flow-through setup to perform mild organosolv extraction (140 °C), focusing on the influence of fractionation time on lignin extraction efficiency, purity, and thermal and chemical properties. They reported an increase in purity over time, an initial increase in extraction rate in the first hour and an increase in condensed structure in the latter fractions [38]. The use of flow-through setups has also been successfully applied into reductive catalytic fractionation [19,21,23], which shows great potential as a marked increase in monomer yields is reported for these setups [19].

We previously reported excellent retention of the β-O-4 linking motifs during mild organosolv extractions in batch setups at 80 °C [45,46], but at these conditions (80:20 EtOH/H_2_O, 0.24 M HCl, 5 h) the extraction yields were relatively low (12% extraction efficiency). Upon increasing the temperature to 120 °C, the extraction efficiency significantly increased, but, in most cases, this came at the expense of a significant drop in the number of β-O-4 linking motifs. Hence, this study was performed to see if an increase in extraction efficiency could be achieved without a loss of β-O-4 content using a flow-through setup instead of a batch reactor. This was done by performing mild organosolv extractions at 120 °C in a developed flow-through setup for 5 h and optimizing the system by testing different acid concentrations, alcohol/water ratios, and alcohols with regards to extraction efficiency and structural quality. The optimized conditions were then tested for the extraction of lignin from other different lignocellulosic feedstocks. The results were compared to batch extractions with the same conditions. It is expected that the results from this paper can serve for the further development of efficient extraction setups and integrated depolymerization setups such as applied in lignin-first biorefinery approaches.

## 2. Materials and Methods

### 2.1. Materials

All commercially available chemicals were used as received. Walnut shell powder was bought from Brambleberry (Bellingham, WA, USA). A pretreated Norway spruce/poplar mixture was provided by Industriewater Eerbeek in an ethanol solution. The ethanol was removed in vacuo and the dry flakes were used as raw material. Cedar and beech wood were obtained at a local wood shop (Dikhout, Groningen). In the extractions, in which the influence of the catalyst was tested manually, ground walnut powder was used [46]. Extractives were removed from walnut shell powder, cedar wood, and beech wood by a 2 h toluene extraction at reflux conditions. The toluene was removed by filtration and the feedstocks were dried in a vacuum oven for 16 h at 70 °C.

### 2.2. Flow-Through Organosolv Extraction

The flow-through reactor was loaded with biomass (20.0 g) and the remaining volume was filled with SiO_2_-fused granules to ensure a constant flow profile. The granules, biomass, and glass filters were separated with quartz wool plugs. The solvent enters the reactor from the bottom and leaves the reactor at the top. The system was initially flushed with the extraction solvent at room temperature. After the reactor had filled-up with liquid, the reactor was allowed to reach the desired pressure (6 bar) using the backpressure valve. Once the desired pressure was reached, the system was heated to the desired temperature. The start of the experiment (t = 0) was set as the point when the temperature reached 80 °C, with the exception of the experiments with different acid concentrations which took the reached temperature of 120 °C as t = 0. The standard conditions used in the experiments were: T = 120 °C; p = 6 bar, φ_m_ = 1 g/min (target flow), [H_2_SO_4_) = 180 mM, and a fractionation time of 5 h. These conditions were based on preliminary experiments and earlier work [46]. The increased pressure was required to keep the ethanol in the liquid phase at 120 °C.

The liquid leaving the reactor was collected in one-hour fractions and the amount of solvent leaving the reactor was monitored every 15 min to ensure that the target flow was maintained. The solvent of the collected fractions was removed in vacuo. The obtained solids were re-dissolved in a minimal amount of acetone (5–10 mL) and precipitated in water (200–300 mL). The addition of aqueous Na_2_SO_4_ was required to flocculate lignin obtained with n-butanol extractions. The lignin was obtained by filtration and air-dried overnight. This lignin was used to determine the extraction efficiency with a correction based on alcohol incorporation measured by HSQC NMR analysis (see Section 2.4 and calculations in Appendix A). By combining the weight fractions with the other analytical data, the averaged lignin characteristics for the complete extraction run could be calculated (Appendix B).

### 2.3. Batch Organosolv Extraction

Comparative batch extractions were performed in a reactor (190 mL) filled with biomass (10.0 g) and extraction solvent (100 mL). The extractions were performed for 5 h at the chosen conditions, during which the pressure in the reactor was monitored with a pressure indicator fitted to the reactor. The organosolv liquor was filtered to remove the remaining solids. The reactor was washed with the applied alcohol (2 × 25 mL) which was added to the organosolv liquor to ensure that all products were removed. The work-up was identical to the lignin obtained by flow-through organosolv extraction.

### 2.4. D Heteronuclear Single Quantum Coherence (HSQC) Spectroscopy

Structural analysis was performed by HSQC NMR spectroscopy. For analysis, 60 mg of lignin was dissolved in 0.7 mL of d_6_-acetone. A few drops of D_2_O were added to ensure complete solubility of the lignin. The NMR analysis was performed on a Bruker Ascend™ Neo 600 using the following parameters: (F2 = 11 to −1 ppm), (F1 = 160 to −10 ppm), nt = 4, ni = 512, d1 = 1.5, CNST [2] = 145, and pulse sequence hsqcetgpsi2. Analysis was performed with MestReNova. The obtained values for the detectable linking motifs (β-O-4, β′-O-4, β-β, and β-5) were divided by a factor of 1.3 as the HSQC measurements overestimate these values as was shown in previous work [46].

### 2.5. Gel Permeation Chromatography (GPC)

Molecular weight was determined by gel permeation chromatography (GPC) analysis. Analysis was carried out on a Hewlett Packard 1100 series THF-GPC. Lignin samples (10 mg) were dissolved in THF (1 mL) with toluene as a flow marker. Prior to analysis, the samples were filtered (0.45 μm PTFE syringe filter). Analysis was performed with PSS WinGPC UniChrom.

### 2.6. Biomass Analysis

Determination of the carbohydrate and lignin content was performed on the biomass sources and a selection of the solid organosolv residues, following the laboratory analytical procedure issued by the National Renewable energy laboratory (NREL) [47]. Dried biomass samples of 0.3 g were added to a pressure tube together with 3 mL of 72% sulfuric acid and mixed. The pressure tubes were placed in a water bath and incubated at 30 °C for 60 minutes. Every 10 min, the samples were stirred. After completion of the first step, the samples were diluted to 4% sulfuric acid by adding 84 mL of deionized water. The Teflon caps were screwed securely on the pressure tubes and the diluted acid was mixed by inverting of the tubes. The tubes were added to an autoclave rack and placed in the autoclave. The samples were heated at 121 °C for 60 min. After completion, the samples were allowed to slowly cool to room temperature.

The acid insoluble lignin (AIL) was determined by filtering the hydrolysis liquor over a filtering crucible. After the filtration, the solids were washed with deionized water. The filtering crucibles were dried in an oven for a minimum of four hours, after which the weight of the solid residue was recorded. The acid soluble lignin (ASL) was determined by UV–Vis spectrophotometry, using a background of 4% sulfuric acid in deionized water. The samples were diluted in order to obtain an absorbance range of 0.7–1.0 at wavelength 240 nm.

## 3. Results and Discussion

### 3.1. Flow-Through Setup and Optimization of Extraction Conditions

For the development of mild organosolv extraction, a flow-through setup was built for the semi-continuous extraction of lignin from biomass (Figure 2). In this setup, the extraction solvent was pumped at a rate of 1 g/min through a heated oven containing a 100 mL cylindrical reactor filled with lignocellulosic biomass in which lignin extraction of the stationary biomass source took place. It was envisioned that, due to the much shorter residence time of the extracted lignin (~1 h), the β-O-4 content should remain high due to the limited chance for these fragments to engage in condensation reactions. This should allow for the combination of good extraction efficiency and high structural quality of the lignin.

Initial conditions were selected that were reminiscent to our earlier reported mild ethanosolv batch extractions (80:20 EtOH/H_2_O, 5 h, 120 °C) in which 38% extraction efficiency was obtained with HCl as acid and 48% extraction efficiency with H_2_SO_4_ as acid [46]. The higher extraction efficiency with H_2_SO_4_ came at the expense of undesired lower β-O-4 content (74 linking motifs per 100 C9 units with HCl and 35 linking motifs per 100 C9 units with H_2_SO_4_). Although initial tests were performed with HCl, its use was excluded for the flow-through setup due to the corrosion of some of the filters in the setup. The H_2_SO_4_ was thus selected for the remainder of this work. A starting concentration of 0.18 M H_2_SO_4_ was selected for the first run (Appendix A). Figure 3A shows that, under the selected conditions, the majority of the lignin was extracted in the first three hours, with the peak extraction in the second hour with a lignin concentration of over 25 mg/g solvent (Figure 3A). The initial increase in extraction efficiency is in correspondence with the reported results of Labbé et al. [38]. The increase from the first to the second hour is likely caused by an initial lag-phase of the extraction due to the heating profile of the reactor. A total extraction efficiency of 52% was obtained after 5 h extraction time, which is slightly better compared to similar batch extractions [46]. Analysis of the lignins obtained after every hour revealed that the lignin of each fraction was high in total β-O-4 content (60–65 β-O-4 linking motifs per 100 C9 units) with a small drop in the last fraction (55 β-O-4 linking motifs, Figure 3B). This is in sharp contrast to the number of β-O-4 linking motifs that were obtained in the previously reported batch experiments, which were performed with a slightly lower acid concentration (35 β-O-4 linking motifs). This shows that the shorter residence time of the extracted lignin indeed reduces the influence of post-extraction reactions like condensation reactions. Longer exposure of the lignin trapped in the biomass did lead to some condensation as evidenced from the third fraction onwards. Nevertheless, this amount of β-O-4 linking motifs remained high for the average lignin obtained from the whole extraction run (55 β-O-4 linking motifs) accompanied with minor amounts of condensed material (1.5% S_condensed_). This was evidenced by the appearance of a distinguishable signal in the 2D HSQC spectra (δ_C_/δ_H_ 107/6.55 ppm), which was assigned to the condensation of the S-unit [48]. The intensity of this signal was used to calculate the amount of condensation and was expressed as the amount of S unit condensation relative to the total amount of S units. A deeper analysis of the type of β-O-4 linkages showed an increase in α-ethoxylated (β′-O-4) units in the first three hours, after which this remained constant. The H/G/S ratio also showed a change over time with the first fraction being evenly distributed in S and G units, after which it turned gradually to predominantly S units (Figure 3C). The GPC analysis showed an increase in molecular weight in the latter fractions, with a small decrease in the last fraction (Figure 3D) which was attributed to depolymerization and the subsequent extraction of smaller condensed fragments. Additionally, a second run under the same reaction conditions showed good reproducibility, yielding the same extraction efficiency (52%) and only small deviations in the β-O-4 linkages which only dropped a bit faster (Appendix A). This initial experiment showed the great potential of this setup, as it combines good extraction efficiency with excellent retention of the lignin structure and, thus, gave us incentive to study the use of this setup further.

Firstly, the influence of the acid concentration on the extraction efficiency and lignin properties was examined. As the result with 0.18 M H_2_SO_4_ was already very good, it was decided to start with some small changes in acid concentration (0.12 M and 0.24 M H_2_SO_4_, Appendix A). With 0.24 M H_2_SO_4_, comparable extraction efficiency was obtained after 5 h, but in the first hours, extraction occurred with smaller increments for the extraction with 0.24 M H_2_SO_4_ (Figure 4A). It should be noted that during this experiment, from the second hour onwards, the flow was only half of the target flow, which could explain part of the difference in the extraction rate and can be compensated for by plotting the result as a function of solvent used (Appendix A). The structural quality was similar in the first three hours, after which a sharper decrease in structural quality was observed for the extraction with 0.24 M H_2_SO_4_ (Figure 4B). The extraction with 0.12 M H_2_SO_4_ gave a more gradual extraction profile (Figure 4A), but the total extraction efficiency was slightly lower compared to the extractions with a higher acid concentration. The total β-O-4 content (64 linking motifs) was markedly higher for the extraction with 0.12 M H_2_SO_4_, compared to the extractions with higher acid concentrations (55 linking motifs for 0.24 M H_2_SO_4_, Figure 4B). A total β-O-4 content of 65–70 linking motifs was obtained in the middle fractions (2–4 h) of this extraction and an average β-O-4 content of 64 linking motifs. This is sharp contrast with the 35 β-O-4 linking motifs that we reported for the batch extraction with identical reaction conditions [46], showing the superior performance of this extraction setup in terms of structure preservation, while showing only a small decrease in extraction efficiency. These results clearly show the influence of the acid on lignin fractionation and depending on the requirements (rapid delignification or obtaining lignin of high structural quality), a different acid concentration can be the preferred option.

As it was previously reported that the alcohol/water ratio can strongly influence the extraction efficiency [9,11,30,49], we decided to explore this next (Appendix A). Figure 5A shows that at higher ethanol concentrations, better extraction efficiency was observed, with the 80:20 and 95:5 EtOH/H_2_O ratio being superior to the 50:50 EtOH/H_2_O ratio. The best result was a 57% extraction efficiency at a 95:5 EtOH/H_2_O ratio after 5 h of extraction time. This result is in contrast with most research in batch systems, in which delignification decreased at ethanol fractions above 80% [9,11,49]. This could be caused by a more efficient hydrolysis in the flow-through setup as fresh solvent mixture, and, in particular, acid is constantly added to the system. The structural quality of lignin obtained with the 95:5 ratio showed a sharp drop after 2 h extraction, going down to 32 β-O-4 linking motifs in the third fraction and only 10 β-O-4 linking motifs in the fourth and fifth fractions (Figure 5B). As the majority of the lignin was extracted in the first 2 h (35% extraction efficiency with an average β-O-4 content of 56 linking motifs after 2 h), the total quality did not show a significant decrease (average β-O-4 content of 44 linking motifs after 5 h). An 80:20 EtOH/H_2_O ratio resulted in lignin with the highest β-O-4 content (62 linking motifs). With a 50:50 EtOH/H_2_O ratio, lignin with a higher β-O-4 content was obtained compared to the 95:5 ratio, but it trailed the 80:20 ratio in terms of both extraction efficiency (52% versus 36%) and structural quality (62 versus 52 β-O-4 linking motifs).

As the use of n-butanol (nBuOH) has been reported to improve the extraction efficiency [14,45], water mixtures of both nBuOH and n-propanol (nPrOH) were tested in an 80:20 alcohol:water ratio in the flow-through setup and compared to EtOH (Figure 6, Appendix A). With nPrOH, an extraction profile similar to EtOH was obtained with a slightly higher overall extraction efficiency (57% versus 53%, Figure 6A) and an overall structural quality that was also a bit higher (66 versus 62 β-O-4 linking motifs, Figure 6B). The main difference of the lignin obtained by EtOH and nPrOH was the weight average molecular weight which was significantly higher for the last three fractions obtained by nPrOH (5000–6000 Da, Appendix A) compared to EtOH (2500–3000 Da Appendix A). When nBuOH was used as solvent, a practical problem did arise at room temperature as an 80:20 ratio nBuOH and water are not fully miscible, causing a non-homogeneous solvent influx that led to undesired extraction profiles. In order to obtain complete mixing, a small amount of 1,4-dioxane was added to the solvent system (80:15:5 nBuOH/H_2_O/1,4-dioxane). Using this mixture, an increase in extraction efficiency (74%) was observed but at the expense of a decrease in quality in the latter fractions (41 β-O-4 linking motifs in the third fraction and 32 β-O-4 linking motifs in the fourth fraction and 15%–20% condensation). High extraction efficiencies have been reported for nBuOH but with comparable structural quality as lignin obtained by ethanosolv extractions [14,50]. In order to determine if the added 1,4-dioxane had a negative influence on the lignin properties, control experiments were performed with 1,4-dioxane added to the EtOH/H_2_O and nPrOH/H_2_O solvent mixtures (Appendix A). Only a minor increase in extraction efficiency was observed compared to the runs without 1,4-dioxane (54% versus 53% for EtOH and 59% versus 57% with nPrOH, Figure 6A). However, the structural quality was significantly worse when 1,4-dioxane was used, especially for nPrOH as the β-O-4 content showed a decrease from 66 linking motifs to 55 linking motifs for nPrOH and the degree of condensation increased from 2% to 18% in the latter fractions. This shows that the use of nBuOH is superior over shorter chain alcohols in terms of extraction efficiency, but that the use of 1,4-dioxane has a negative influence on the structural quality.

### 3.2. Increased Reactor Loading and Other Feedstocks

For industrial implementation, it is important that such technology is scalable and applicable on other, possibly mixed, feedstocks that come from agricultural residues. We were therefore eager to test a fully loaded reactor in our flow-through setup (40 grams instead of 20 grams of walnut shell powder) while maintaining the same flow rate. A 95:5 EtOH/H_2_O ratio was selected to reach the maximum extraction efficiency and the extraction solvent was collected per 30 min time period (Appendix A). To our surprise, the extraction efficiency showed a significant increase when the feedstock was doubled (82% instead of 57%, Figure 7A), which can be explained by the lower percentile losses of lignin during the workup at larger scale. This was confirmed by determination of the lignin in the residual material by the Klason method which showed similar delignification results for both experiments (77% and 79%). This also indicates that the actual lignin removal for most experiments is actually higher than the provided data based on wt % of recovered lignin suggests. The higher recovery of lignin from the 40 g experiment also reveals that the solvation capacity for lignin is much higher compared to the extraction with lower reactor loading as in this experiment the outflow contained more than double the lignin concentration (49 mg/g compared to 18 mg/g). This significant improvement in solvent efficiency is promising for further development, as this tackles the high solvent consumption which is one of the drawbacks of the flow-through setup compared to the batch setup. Analysis of the lignin characteristics showed similar structural quality as well as other properties such as *M_w_*, linkage distribution, and H/G/S ratio for both extractions (Appendix A), showing no negative effect for full loading and longer contact time. These results indicate that solvent consumption use per gram of extracted lignin can be further improved.

Additionally, different biomass sources (i.e., spruce/poplar mixture, cedar wood, and beech wood) were tested as feedstock under these selected ethanosolv conditions (20 g, 95:5 EtOH/H_2_O, 0.18 M H_2_SO_4_, Figure 8 and Appendix A). With beech wood, good overall delignification was achieved (73%), whilst for cedar wood and spruce/poplar far less efficient delignification was achieved (29% and 20%, respectively). Analysis of the obtained lignins also showed large deviations in structural quality. Lignin obtained from spruce/poplar contained a much higher overall β-O-4 content (57 linking motifs) compared to lignin obtained from walnut shell (44 linking motifs), beech wood (43 linking motifs), and cedar wood (35 linking motifs). The latter three biomass sources showed a significant decrease in β-O-4 content in the last two fractions, whereas the structural quality of lignin from spruce/poplar remained high throughout the entire extraction. This indicates that the conditions require tuning for each feedstock. Similar results were also found in earlier work in batch setups [46].

### 3.3. Flow-Through Extractions Compared to Batch Extractions

To clearly demonstrate the advantage of using the flow-through setup, most of the extractions discussed above were repeated in a batch setup using the same conditions and the results were compared (Figure 9). The flow-through setup proved to be superior for all these comparative experiments in terms of both extraction efficiency as well as quality of the obtained lignin. For most extractions, an increase in the range of 20% in extraction efficiency was achieved. Additionally, the flow-through setup provided lignin with a β-O-4 content in the range of 45–60 β-O-4 linking motifs, whereas in the batch system the β-O-4 content decreased to 8–30 β-O-4 linking motifs for the extractions performed with over 80% EtOH. This clearly shows that a shorter residence time of extracted lignin in the reactor is key for obtaining high-purity lignin. With nPrOH and nBuOH, the effects were not as profound, as the β-O-4 content was high with both extraction setups, but still an increase in both structural quality and extraction efficiency were observed for the flow-through setup. Furthermore, the degree of condensation was considerably lower for most flow-through extractions as was the increase in molecular weight for the obtained lignin (Appendix A). Analysis of the solvent efficiency (Appendix B and Appendix C) showed that the batch setup was somewhat more efficient after 5 h extraction when EtOH and nPrOH were used as solvent. With nBuOH as solvent, the flow-through setup was the slightly more efficient system. A closer look at the values shows that the extraction efficiency of 2 h extraction with the flow-through setup was, in most cases, comparable to the extraction efficiency of 5 h batch extractions. When taking this into consideration, the flow-through setup had a markedly higher solvent efficiency compared to the batch setup. Thus, overall it can be concluded that the flow-through extraction setup was superior to a conventional batch setup in terms of both extraction efficiency and structural quality and being comparable in terms of solvent efficiency.

## 4. Conclusions

A flow-through setup was developed for the semi-continuous mild organosolv extraction of lignin with acidic alcohol water mixtures. Most of the extractable lignin was removed from the biomass in the first three hours of the experiments. The lignin of the highest structural quality, in terms of β-O-4 content, was obtained during the first part of the extraction. The acid and alcohol concentration, as well as the choice of alcohol, were shown to influence the extraction efficiency and quality of the obtained lignin. Furthermore, it was demonstrated that the increased reactor loading was beneficial for the lignin yield, and this was attributed to a lower amount of lignin loss during work-up while keeping a constant lignin quality which is a promising feature for further upscaling. These results for the different alcoholic solvents and ratios can be utilized for further optimization of lignin-first methodologies, as the influence of the solvent choice has a direct influence on the final phenolic composition [51]. Large variations in the results were observed for different biomass feedstock, indicating that the conditions need to be tuned separately for each feedstock. The comparison between flow-through extractions and batch extractions showed a significant increase in extraction efficiency and a higher structural quality of the lignin obtained by extractions performed in the flow-through setup. These results show the high potential of the flow-through setup for future applications as high-quality lignin is extracted with good yields. The ability to collect lignin in multiple fractions allows for the collection of lignin fractions with a structural quality of up to 70 β-O-4 linking motifs, which has a lot of potential to be applied into reaction pathways towards high-value products. Furthermore, this setup could be implemented into lignin-first methodologies, as recent work has demonstrated the high potential of flow-through reductive catalytic fractionation [19,21].

## Figures and Tables

**Figure 1 polymers-11-01913-f001:**
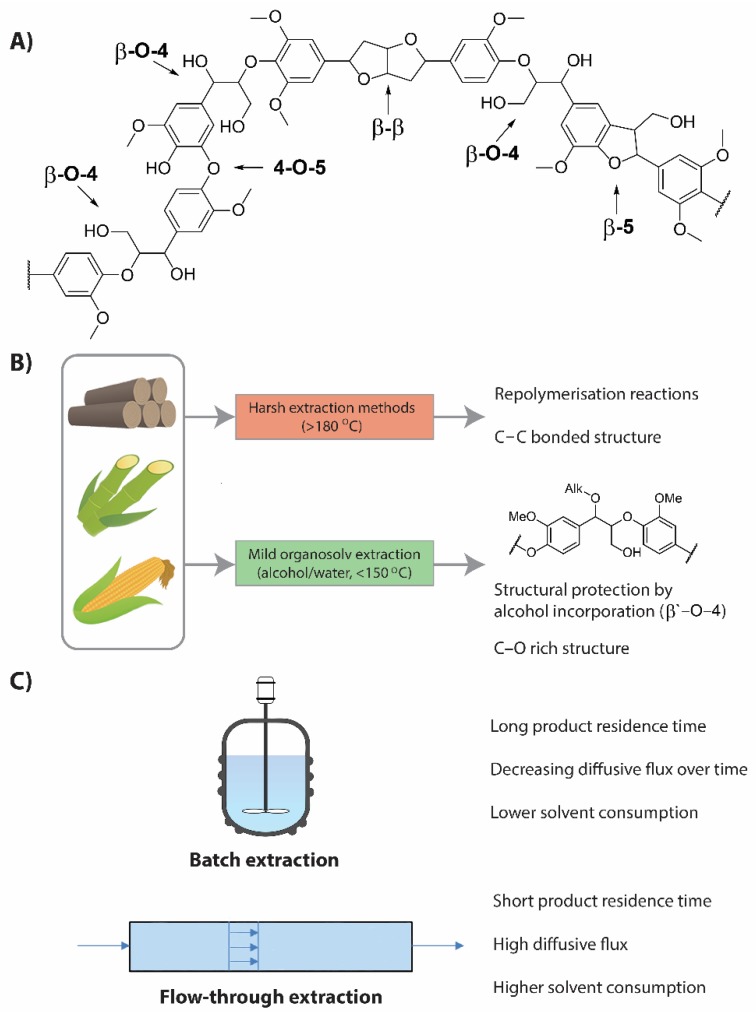
Schematic representation of the lignin structure (**A**); characteristics of lignin obtained by harsh extraction and mild organosolv extraction (**B**); and key characteristics of batch and flow-through extraction setups (**C**).

**Figure 2 polymers-11-01913-f002:**
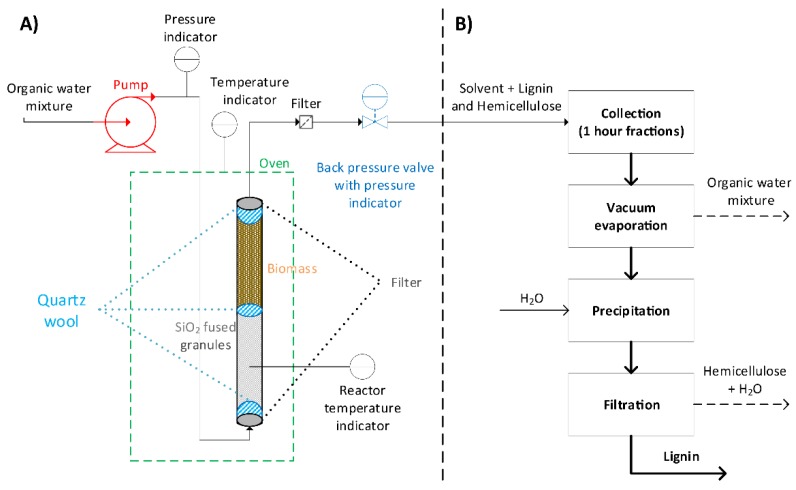
Schematic overview of the extraction procedure, showing (**A**) the developed flow-through setup and (**B**) the workup procedure for lignin isolation.

**Figure 3 polymers-11-01913-f003:**
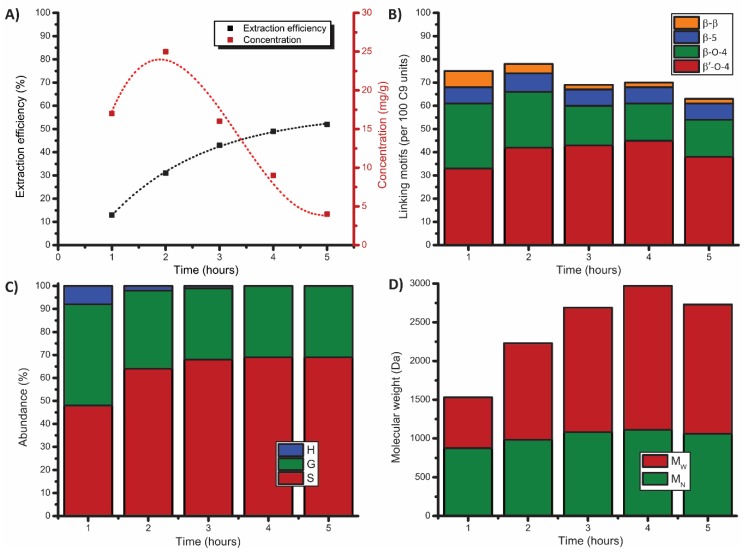
Results of the mild organosolv extraction (80:20 EtOH/H_2_O, 0.18 M H_2_SO_4_, 120 °C, 5 h), showing (**A**) extraction efficiency (corrected for alcohol incorporation as determined by 2D HSQC) and lignin concentration, (**B**) linking motifs distribution and (**C**) H/G/S ratio as determined by 2D HSQC, and (**D**) molecular weight distribution as determined by GPC (THF).

**Figure 4 polymers-11-01913-f004:**
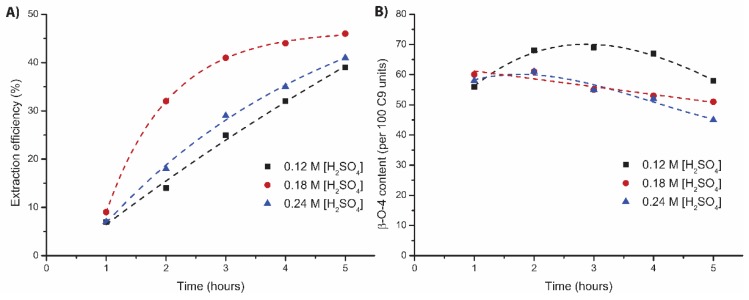
Influence of the different H_2_SO_4_ concentrations on the mild organosolv extraction * (80:20 EtOH/H_2_O, 120 °C, 5 h) on (**A**) extraction efficiency (corrected for alcohol incorporation as determined by 2D HSQC) and (**B**) total β-O-4 content as determined by 2D HSQC. * Different walnut source than starting point experiment.

**Figure 5 polymers-11-01913-f005:**
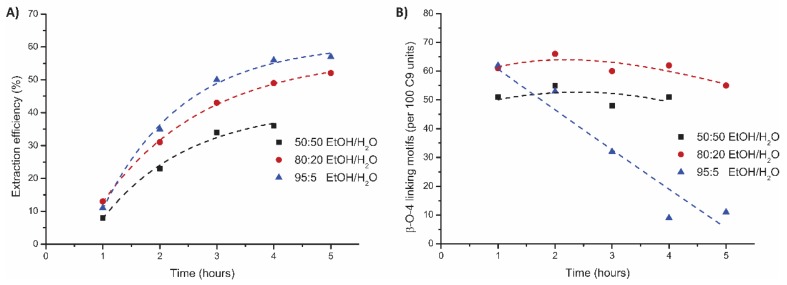
Influence of the different EtOH/H_2_O on the mild organosolv extraction * (0.18 M H_2_SO_4_, 120 °C, 5 h) on (**A**) extraction efficiency (corrected for alcohol incorporation as determined by 2D HSQC) and (**B**) total β-O-4 content as determined by 2D HSQC. * The 50:50 EtOH/H_2_O experiment was stopped after 4 h due to the reactor clogging.

**Figure 6 polymers-11-01913-f006:**
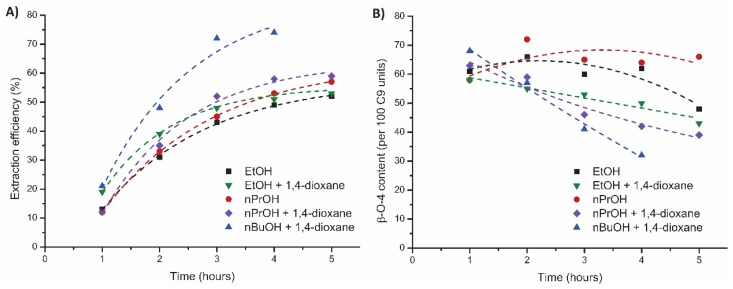
Influence of alcohol on the mild organosolv extraction (80:20 alcohol/H_2_O *, 0.18 M H_2_SO_4_, 120 °C, 5 h) on (**A**) extraction efficiency (corrected for alcohol incorporation as determined by 2D HSQC) and (**B**) total β-O-4 content as determined by 2D HSQC. *W hen 1,4-dioxane was used, an 80:15:5 alcohol/H_2_O/1,4-dioxane ratio was applied.

**Figure 7 polymers-11-01913-f007:**
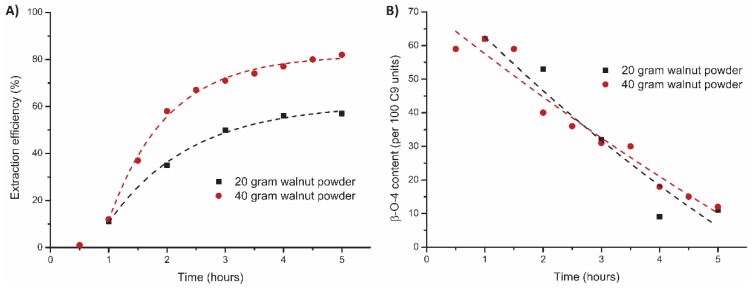
Influence of reactor loading on the mild organosolv extraction (95:5 EtOH/H_2_O, 0.18 M H_2_SO_4_, 120 °C, 5 h) on (**A**) extraction efficiency (corrected for alcohol incorporation as determined by 2D HSQC) and (**B**) total β-O-4 content as determined by 2D HSQC.

**Figure 8 polymers-11-01913-f008:**
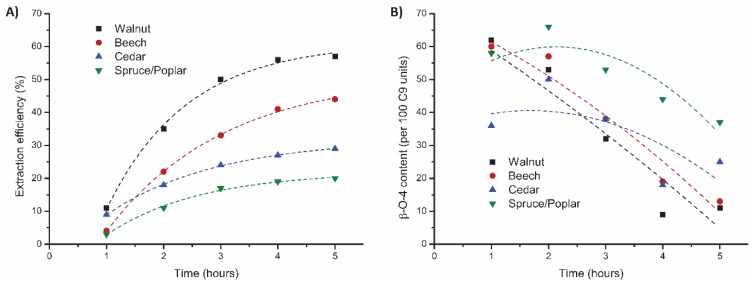
Influence of the feedstock on the mild organosolv extraction (95:5 EtOH/H_2_O, 0.18 M H_2_SO_4_, 120 °C, 5 h) on (**A**) extraction efficiency (corrected for alcohol incorporation as determined by 2D HSQC) and (**B**) total β-O-4 content as determined by 2D HSQC.

**Figure 9 polymers-11-01913-f009:**
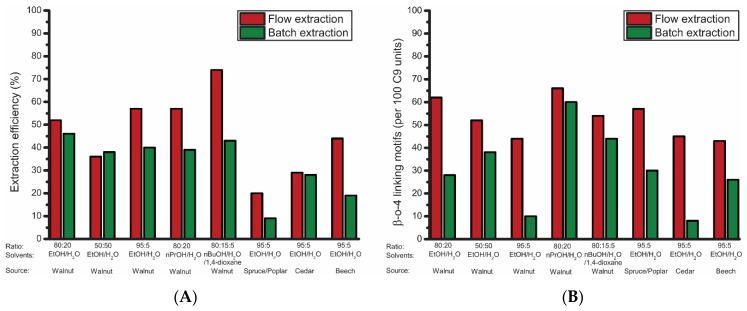
Comparison of flow-through and batch extractions, showing the influence of the reactor setup on (**A**) extraction efficiency (corrected for alcohol incorporation) and (**B**) total β-O-4 content as determined by 2D HSQC.

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
