# Peer review of "Efficient Mild Organosolv Lignin Extraction in a Flow-Through Setup Yielding Lignin with High β-O-4 Content"

_polymers, 2019, doi:10.3390/polym11121913_

Round 1

Reviewer 1 Report

The manuscript by Deuss and coworkers entitled "Efficient mild organosolv lignin extraction in a flow-through setup yielding lignin with high β-O-4 content" describe how flow-through technology affects the lignin as compared to batch reactions. 

This is a thorough study where a lot of different parameters have been evaluated with respect to different qualities of the lignin such as beta-O-4 content and molecular weight.

My only concern, and this is surely a minor one, is the time slots for evaluation. 1 hour sampling time is quite long. Even if the efficiency is measured to be higher at 2 hours, shorter times could give additional information. However, this can be discussed in a follow-up study.

This reviewer support publication!

Author Response

Author response to Reviewer 1

The manuscript by Deuss and coworkers entitled "Efficient mild organosolv lignin extraction in a flow-through setup yielding lignin with high β-O-4 content" describe how flow-through technology affects the lignin as compared to batch reactions.

This is a thorough study where a lot of different parameters have been evaluated with respect to different qualities of the lignin such as beta-O-4 content and molecular weight.

Response: Thanks you for the kind comments regarding our research

My only concern, and this is surely a minor one, is the time slots for evaluation. 1 hour sampling time is quite long. Even if the efficiency is measured to be higher at 2 hours, shorter times could give additional information. However, this can be discussed in a follow-up study.

Response: We agree that additional time can provide additional information. However, in this study we are limited by the amount of sample needed to get high quality HSQC-NMR spectra for the quantification of the lignin linkage motifs. To ensure we collect enough lignin sample 1 hour sampling times were necessary, especially in experiments that provide lower yields. Already when we double the biomass loading in the reactor we were able to move to 30 min sampling time. Indeed in the future in larger setups more time data points can be obtained providing further insight.

This reviewer support publication!

Reviewer 2 Report

This paper is definitely of interest to the field by going from usually applied batch reactors for organosolv delignification to a flow-through setup. Thus, not only scientifically additional know-how is generated considering the processes occurring, but also a step is set towards a more industrial relevant approach. Furthermore, it is a decent paper and should therefore be published. At the same time, the authors could  have substantially improved their paper by using a more logical experimental design as well as giving the potential industrial application a better thought.

Experimental design. The experiments performed as presented in Appendix A do not show any logical order of process conditions tested. Using a Design of Experiments or another systematic approach would have improved the clarity of presentation to the reader as well as allow for statistical determination of optimum process conditions.

Industrial applicability. For industrial application, the residence time should be shortened. The authors now present a concept that even uses a longer residence time (up to 5 hr) than what is typically applied in batch systems. Why did the authors not look into more relevant process conditions that allow for delignification in let's say <1hr? Same holds for the low product concentration. Thus, more relevant results could have been presented.

Section 3.2 The authors should not use the term scalability here. What the authors tested here is increase of biomass loading / higher product concentrations.

Other. Introduction, page 2, line 53-44. Using alcohol can indeed play this function. However, it is not needed to use an alcohol to get high B-O-4 lignin. Other studies have shown that it is also possible to do this by using mild organosolv conditions with other solvents. Could the authors elaborate on this?

Page 4, lines 118-123 are unclear. Please check.

Page 5, section 3.1. A substantial part of this actually belongs to the methods section.

Page 9, lines 316 and further. I am bit afraid that if this is the case all data presented are influenced by the work-up method. In other words to what extent are effects that the authors discuss related to the actual delignification? Not only in the sense of an underestimation as stated on page 10, lines 320-322, but also in the sense of variations in the data. Please comment.

Author Response

Author response to Reviewer 2

This paper is definitely of interest to the field by going from usually applied batch reactors for organosolv delignification to a flow-through setup. Thus, not only scientifically additional know-how is generated considering the processes occurring, but also a step is set towards a more industrial relevant approach. Furthermore, it is a decent paper and should therefore be published. At the same time, the authors could  have substantially improved their paper by using a more logical experimental design as well as giving the potential industrial application a better thought.

Response: We thank the reviewer for the kind comments regarding our research, below the point regarding experimental design and potential industrial application are addressed.

Experimental design. The experiments performed as presented in Appendix A do not show any logical order of process conditions tested. Using a Design of Experiments or another systematic approach would have improved the clarity of presentation to the reader as well as allow for statistical determination of optimum process conditions.

Response: We agree with the reviewer that the experimental design could have been better. This study was for us an exploration in the potential for a flow-through extraction setup and to gain insight in what parameter are important for balancing yield and structural integrity of the extracted lignin. The aim was not to perform full optimization, also as it depends of the application and the overall process if higher yield or higher lignin linkage content is desired. Furthermore, as the experiment with different feedstock clearly demonstrate optimization for each substrate will be necessary. We hope that the data provided here can guide more focused experiments in which there a clear lignin specification requirement starting from a set biomass feedstock.

Industrial applicability. For industrial application, the residence time should be shortened. The authors now present a concept that even uses a longer residence time (up to 5 hr) than what is typically applied in batch systems. Why did the authors not look into more relevant process conditions that allow for delignification in let's say <1hr? Same holds for the low product concentration. Thus, more relevant results could have been presented.

Response: We agree with the reviewer that long residence times combined with the low concentrations are indeed one of the concerns for industrial application and are for us a focus for further research. There are indeed methods out there that can reach high delignification at short extraction times. However, these typically yield heavily structurally altered lignin (mostly highly condensed lignin with low amount of β-O-4 linkages) or are not compatible with our setup. We do want to highlight that if one looks at the extraction efficiency after 2 hours for entry 11 in appendix A already 58% delignification is observed based on recovered solid lignin. Tuning the solvent system and further reactor design are ways we hope to improve on our current process.

Section 3.2 The authors should not use the term scalability here. What the authors tested here is increase of biomass loading / higher product concentrations.

Response: We thank the reviewer for this suggestion and  have changed the text of the manuscript accordingly.

“An experiment with higher reactor loading” in the abstract. “3.2 Increased reactor loading and other feedstocks”, “We were therefore eager to test a fully loaded reactor In our flow-through setup”, “The higher recovery of lignin from the 40 gram experiment also reveals that the solvation capacity for lignin is much higher compared to the extraction with lower reactor loading” and “showing no negative effect for full loading and longer contact time. These results indicate that solvent consumption use per gram of extracted lignin can be further improved.” in section 3.2

Other. Introduction, page 2, line 53-44. Using alcohol can indeed play this function. However, it is not needed to use an alcohol to get high B-O-4 lignin. Other studies have shown that it is also possible to do this by using mild organosolv conditions with other solvents. Could the authors elaborate on this?

Response: Indeed there are other solvent mixtures that allow for the extraction of lignin with relatively high β-O-4. For example acetone, γ-valerolactone and methyl isobutyl ketone are also used and show high β-O-4 content. However, the reported values are somewhat lower than the values for alcohols. We have also experienced this with our own extractions.  Alcohols in our experience especially in combination with the lignocellulosic feedstock we have worked with allow for more robust extraction of lignin with high β-O-4 content, due to the provided protection by alcohol incorporation into the structure. As references to other solvents were lacking, we added references to papers in which the aforementioned 3 solvents are used. We also rewrote the sentence in line 53 somewhat.

“with different organic solvents, which allows for obtaining lignin with relatively high β-O-4 content. [25–27] These extractions are performed”

Fang, W.; Sixta, H. Advanced Biorefinery based on the Fractionation of Biomass in g -Valerolactone and Water. 2015, 73–76. Luo, H.; Abu-omar, M.M. Lignin extraction and catalytic upgrading from genetically modified poplar. Green Chem. 2018, 20, 745–753. Bozell, J.J. Optimization of Component Yields and Thermal Properties by Organosolv Fractionation of Loblolly Pine ( Pinus taeda ) Using Response Surface Design. Bioenergy Res. 2018, 11, 652–664

Page 4, lines 118-123 are unclear. Please check.

Response: We have adjusted the text to be clearer:

“The solvent enters the reactor from the bottom and leaves the reactor at the top. The system was initially flushed with the extraction solvent at room temperature. After the reactor had filled-up with liquid the reactor was allowed to reach the desired pressure (6 bar) using the backpressure valve. Once the desired pressure was reached, the system was heated to the desired temperature. The start of the experiment (t = 0) was set as the point when the temperature reached 80 ⁰C, with the exception of the experiments with different acid concentrations, which took the reached temperature of 120 ˚C as t = 0.”

Page 5, section 3.1. A substantial part of this actually belongs to the methods section.

Response: We had a critical look and removed overlapping text that was not essential for the overall story.

“At the start of reaction the solvent was added to the system at room temperature until a steady flow at the desired pressure (6 bar) was obtained. Next, the system was heated to the desired temperature (120 ˚C). The point at which the temperature reached 80 ˚C was chosen as t=0 of the extraction and collection of the liquid outflow was started. Typically, the liquid was collected per 1 hour time period and the lignin was recovered and analysed” was removed and “By combining the weight fractions with the other analytical data also averaged lignin characteristics for the complete extraction run could be calculated (Appendix A).” was moved to section 2.2.

Page 9, lines 316 and further. I am bit afraid that if this is the case all data presented are influenced by the work-up method. In other words to what extent are effects that the authors discuss related to the actual delignification? Not only in the sense of an underestimation as stated on page 10, lines 320-322, but also in the sense of variations in the data. Please comment.

Response: There will be some influence of the work-up procedure, but this workup is necessary to recover the lignin in a form that allows us to perform the required analysis for measuring the desired properties by HSQC-NMR. Although, there are solids lost we are convinced that the recovered fraction is representative of the total extracted lignin. We are encourage by: Firstly, that clear trends can be observed from our experiment and secondly, when compared to batch, clear improvements in yield can still be observed, while in batch less losses due to the work-up are expected (1 work-up instead of the average of 5 for the flow).

Reviewer 3 Report

This paper introuduced one flow-through setup for the semi-continuous mild organosolv extraction of lignin with acidic alcohol water mixtures.  The research background, analysis methods and results were clearly presented. The authors are encouraged to address the following comments:

1.  In the Abstract, for the " The best structural quality". How to define the "best"? They are suggested to be reflected in the Abstract, as well as the conclusion part.

2. In the first graph of "Introduction", more references should be cited for the previous studies. Such as "One of the developed mild fractionation methods is mild organosolv extraction..." and " Recent kinetic analysis on delignification showed that the rate determining step of the delignification process is the diffusion of dissolved lignin molecules within the cellulose layers".

3. In line 43-45, the expression of the sentence "Obviously, these indicators are not scientific and cannot meet the actual needs of road engineering, so they can not guide the actual construction and be used as quality evaluation." is too absolute.

4. I am not quite familair with the setup developed in this study. The authors should elaborate how to setup and use this device. I cannot imagine how to get the results, and does it work as it is very specific device, and not specified in any standard.

5.  The "Conclusions" is a little bit extensive. And it looks like "Record and analysis of resuslts", it should be high concluded.

6. Can you show the standard deviation for each point of test? As you show only one point for each test, there would not represent precision and bias of using the devices.

Author Response

Author response to Reviewer 3

This paper introuduced one flow-through setup for the semi-continuous mild organosolv extraction of lignin with acidic alcohol water mixtures.  The research background, analysis methods and results were clearly presented.

Response: We thank the reviewer for these nice comments

The authors are encouraged to address the following comments:

In the Abstract, for the " The best structural quality". How to define the "best"? They are suggested to be reflected in the Abstract, as well as the conclusion part.

Response: We have changed the discussion to better indicated what is referred to when “best” is used:

“High structural quality” and “the best structural quality, in terms of β-O-4 linking motif content” in the abstract. As well as, “... in terms of β-O-4 content,..” in the conclusions part.

In the first graph of "Introduction", more references should be cited for the previous studies. Such as "One of the developed mild fractionation methods is mild organosolv extraction..." and " Recent kinetic analysis on delignification showed that the rate determining step of the delignification process is the diffusion of dissolved lignin molecules within the cellulose layers".

Response: We agree with the reviewer that more references should be cited. To the first sentence cited by the referee we added three additional references in which organic solvents other than alcohols are applied.

Fang, W.; Sixta, H. Advanced Biorefinery based on the Fractionation of Biomass in g -Valerolactone and Water. 2015, 73–76. Luo, H.; Abu-omar, M.M. Lignin extraction and catalytic upgrading from genetically modified poplar. Green Chem. 2018, 20, 745–753. Bozell, J.J. Optimization of Component Yields and Thermal Properties by Organosolv Fractionation of Loblolly Pine ( Pinus taeda ) Using Response Surface Design. Bioenergy Res. 2018, 11, 652–664

With regards to the second cited sentence, the following references have been added:

Kumaniaev, I.; Subbotina, E.; Sävmarker, S.J.; Larhed, M.; Galkin, M. V; Samec, J.S.M. Lignin depolymerization to monophenolc compounds in a flow through system. Green Chem. 2017, 19, 5767–5771.

23          Anderson, E.M.; Stone, M.L.; Hülsey, M.J.; Beckham, G.T.; Román-Leshkov, Y. Kinetic Studies of Lignin Solvolysis and Reduction by Reductive Catalytic Fractionation Decoupled in Flow-Through Reactors. ACS Sustain. Chem. Eng. 2018, 6, 7951–7959.

Renders, T.; Van den Bossche, G.; Vangeel, T.; Van Aelst, K.; Sels, B. Reductive catalytic fractionation: state of the art of the lignin-first biorefinery. Curr. Opin. Biotechnol. 2019, 56, 193–201.

In line 43-45, the expression of the sentence "Obviously, these indicators are not scientific and cannot meet the actual needs of road engineering, so they can not guide the actual construction and be used as quality evaluation." is too absolute.

Response: We could not quite find the quote text, but we had a critical look at lines 43-45 and improved the text:

“Most of the applied lignin fractionation methods apply harsh extraction conditions that break a significant amount of β-O-4 linking motif,[9–13] resulting in a more recalcitrant, condensed C-C bonded structure via repolymerisation reactions, hampering the application of the selective depolymerisation methodologies targeting this linking motif.”

I am not quite familair with the setup developed in this study. The authors should elaborate how to setup and use this device. I cannot imagine how to get the results, and does it work as it is very specific device, and not specified in any standard.

Response: We feel our setup is quite well described. We have tried to improve our description of how the setup is operated in section 2.2 according to reviewer 2 comments 5 and 6. Also, to get a better feel for how it looks in practice, we added a picture of the setup in the supporting information (Figure S1).

The "Conclusions" is a little bit extensive. And it looks like "Record and analysis of resuslts", it should be high concluded.

Response: We agree that the conclusions was long as we highlighted some of our best results. It has been rewritten to stress the main findings which greatly reduced the size of the conclusion.

“A flow-through setup was developed for the semi-continuous mild organosolv extraction of lignin with acidic alcohol water mixtures. Most of the extractable lignin was removed from the biomass in the first three hours of the experiments. The lignin of the highest structural quality, in terms of β-O-4 content, was being obtained during the first part of the extraction. The acid and alcohol concentration, as well as the choice of alcohol, were shown to influence the extraction efficiency and quality of the obtained lignin. Furthermore, it was demonstrated that increased reactor loading was beneficial for the lignin yield, as this attributed to a lower amount of lignin loss during work-up, while keeping a constant lignin quality which is a promising feature for further upscaling. These results for the different alcoholic solvents and ratios can be utilized for further optimization of lignin-first methodologies, as the influence of the solvent choice has a direct influence on the final phenolic composition.[48] Large variations in the results were observed for different biomass feedstock, indicating that the conditions need to be tuned separately for each feedstock. The comparison between flow-through extractions and batch extractions show a significant increase in extraction efficiency and a higher structural quality of the lignin obtained by extractions performed in the flow-through setup. These results show the high potential of the flow-through setup for future applications as high quality lignin is extracted with good yields. The ability to collect lignin in multiple fractions allows for the collection of lignin fractions with a structural quality of up to 70 β-O-4 linking motifs, which has a lot of potential to be applied into reaction pathways towards high-value products. Furthermore, this setup could be implemented into lignin-first methodologies, as recent work demonstrated the high potential of flow-through reductive catalytic fractionation.[20,21]

Can you show the standard deviation for each point of test? As you show only one point for each test, there would not represent precision and bias of using the devices.

Response: A re-run of a  EtOH/H2O 80:20, 0.18 M [H2SO4] extraction was performed and the result was added to the supporting information (Table S17). These results show that the extraction is reproducible as the final yield is identical and the properties and trends of the isolated lignin are comparable. The following sentence was added to section 3.1:

“Additionally a second run under the same reaction conditions showed good reproducibility yielding the same extraction efficiency (52%) and only small deviations in the β-O-4 linkages, which only dropped a bit faster (Table S16 and S17).”